# Changes in Postural Control Ability after Wearing Corrective Glasses for Distance in Older Adults and Their Causes

**DOI:** 10.3390/ijerph19116643

**Published:** 2022-05-29

**Authors:** Dong-Sik Yu, Sang-Yeob Kim

**Affiliations:** Department of Optometry, College of Health Science, Kangwon National University, Samcheok-si 25949, Korea; yds@kangwon.ac.kr

**Keywords:** corrective glasses, postural stability, synchronization index, sway power index, TETRAX system

## Abstract

Deterioration of postural control in older adults is unavoidable due to age-associated degeneration of the visual system. Our study objective was therefore to verify whether there is a positive effect on postural control ability after providing older adults, whose postural control function had deteriorated, with corrective glasses to correct refractive errors. Twenty-seven participants, 66 to 90 years of age, were included in this study. Stability index, synchronization index, and sway power index were measured by using the TETRAX balance system and comparatively analyzed before and after wearing the corrective glasses for distance. The stability index after wearing corrective glasses significantly decreased compared to before wearing them (*p* < 0.05). Four synchronization areas, among six, showed a significant increase in the synchronization index after wearing corrective glasses compared to before wearing them (*p* < 0.05). The sway power index significantly decreased in the mid–high and the high frequency after wearing the corrective glasses (*p* < 0.05). Optimal visual information can be obtained through the correction of residual refractive errors, eliciting a positive effect on the overall posture control by improving the sensory integration ability with the somatic nervous system responsible for posture control, maximizing the lower extremity motor function of the elderly.

## 1. Introduction

The visual system refers to an essential sensory organ that enables smooth posture control by continuously providing body position information through the recognition of objects and the surrounding environment [1]. A decrease in postural control ability is unavoidable in older adults due to physiological age-related degenerative changes, as well as deterioration of sensory and cognitive functions. The following have been identified as major factors that increase the risk of postural instability and falls in older adults: the ability of balance and gait control [2], musculoskeletal function [3,4], cardiovascular function [5,6,7], vestibular function [8,9], and somatic function [10,11]. As such, each specialized field is making efforts to prevent fall incidence in older adults by identifying the causes of postural control deterioration in the elderly and proposing the necessary solutions.

Degenerative changes in visual function occur as people age. Deterioration of visual functions, such as visual acuity, stereopsis, and contrast sensitivity, in older adults is known to be a visual factor that increases the risk of falls by reducing postural control ability [12,13,14,15]. Senile cataract is particularly a very common eye disease in the elderly and may be a natural process that occurs with age [16]. Although senile cataract is a common and treatable degenerative eye disease, it results in a nonoptimal vision state (blurred vision) that acts as a negative factor for smooth posture control [17]. According to a study by Jack et al. [18], visual problems were present in 76% of elderly patients hospitalized for fall incidences, and in 79%, vision recovery was possible, by either refractive error recorrecting (40%) or cataract surgery (37%). These results suggest that an appropriate optical correction for residual refractive error alone can have a significant effect in preventing frequent fall accidents in older adults.

Edwards [19] and Paulus et al. [20] studied the relationship between refractive error and postural control and reported a 25–50% increase in body sway when myopic blur was induced by using a spherical lens of +4.00 D to +6.00 D. In a subsequent study by Paulus et al. [21], an average of 25% body sway occurred in real myopic patients wearing corrective glasses ranging from S–3.00 D, compared to S–11.00 D when they were not wearing corrective glasses. Since these previous studies had limitations in experimenting with a single condition of myopic blur, the authors investigated the effects of refractive errors by type on postural control and fall risk in a previous study [22,23,24,25]. As a result, we found that myopic blur and astigmatic blur both had negative effects on postural control [22]. Moreover, the participants reported a decrease in postural stability compared to the fully corrected condition, despite having an average unaided visual acuity of 1.0 or higher, when the hyperopic refractive error condition was induced [23]. Furthermore, in a previous study [24,25], the authors analyzed the postural stability improvement effect by having young adults directly wear fully corrective glasses. The results revealed that postural stability was significantly improved compared to before wearing the corrective glasses, indicating that the optical correction effect was effective immediately after wearing the corrective glasses, and further stabilized after wearing the glasses for 6 h. However, the authors’ previous study [24,25] had a limitation in that it was an experiment conducted on healthy young adults. Moreover, it is deemed necessary to prove whether optical correction would have a positive effect on older adults who have a substantially decreased postural control ability and are more exposed to the risk of falls.

Therefore, in this study, older adults who had not worn corrective glasses for distance within the last year were asked to wear fully corrective glasses. The purpose of this study was to investigate the change in postural control ability and identify which sensory organ involved in postural control was positively affected by refractive error correction when postural control ability was improved, to determine the causal relationship. Based on this, we would like to emphasize the social role of optometrists in improving the postural control ability of older adults, preventing fall accidents, and presenting academic and clinical implications for reference in related specialized fields, such as orthopedic surgery, physical therapy, and occupational therapy.

## 2. Materials and Methods

### 2.1. Participants

This study was conducted on 27 participants (66 to 90 years of age). In the 27 subjects included in this study, 18 subjects had hyperopic refractive error in both eyes, 3 subjects had myopic refractive error in both eyes, and 6 subjects had a combination of hyperopic and myopic refractive error in each eye. The inclusion criteria were as follows: participants who had not, within the last year, worn daily glasses for distance, and participants who could walk independently without aids or assistive devices. Medical interviews confirmed that the participants had no history of glaucoma; macular degeneration; retinal problem caused by diabetes; hypertensive retinal disease; strabismus and experience in surgery or treatment related thereto; and frequent falls or any diseases related to body imbalance, systemic diseases, and medications (neuromuscular and musculoskeletal). All participants had a medical history of senile cataracts in one or both eyes, as diagnosed by their ophthalmologist. Individuals with less severe cataracts who had not yet undergone cataract surgery were selected as participants. In addition, there were no subjects with impaired reading ability or cognitive problems during the visual acuity and refraction tests due to the elderly. Table 1 shows the full-corrective prescription for refractive errors and aided decimal visual acuity in all participants who participated in this study. The purpose and method of the experiment were clearly explained to all the participants (verbally and in writing). The experiments were conducted after obtaining participants’ informed consent.

### 2.2. Measurement Equipment

The change in postural control ability after wearing corrective glasses was evaluated by using the TETRAX interactive postural balance system (Tetrax Portable Multiple System, Tetrax Ltd., Ranmat Gan, Israel) (Figure 1). The TETRAX system consists of four ground reaction force sensors divided into A (left heel), B (front part of right sole), C (right heel), and D (front part of left sole). The four ground reaction force sensors comprehensively analyzed the sway area, length, speed, and center of gravity movement pattern in a static state for 32 s and numerically displayed this information for the experimenter in order to evaluate various body balance abilities [26].

### 2.3. Measurement Factors

The following analyses were performed by using TETRAX [26,27,28].

#### 2.3.1. Stability Index

The stability index calculates the value according to the change in weight carried on the TETRAX ground reaction force sensors and an index that indicates the degree of overall postural stability produced by analyzing the degree of postural sway of the subject. An increase in this index is interpreted as a decrease in the overall postural stability.

#### 2.3.2. Synchronization Index

Synchronization index analysis shows the waveform correlation of vibrations measured in the selected two ground reaction force sensors. The interaction between each area is analyzed for the ground reaction force sensors by measuring the change in weight distribution on the entire left and right foot, as well as the front part and heel of each foot. The ground reaction force areas composed of A, B, C, and D are classified by pairs into AB (entire left foot), CD (entire right foot), AC (left and right heels), BD (front parts of left and right soles), AD (left heel and front part of right sole), and BC (front part of left sole and right heel), to analyze the synchronization ability (Figure 1B). When AB, CD, AD, and BC have a negative (−) value and AC and BD have a positive (+) value, synchronization ability is interpreted as superior. The measurement range of these values is from −1000 to 1000. The absolute value of 700 or higher means that the body balance ability is normal, and the absolute value of lower than 700 means that the synchronization ability between each foot area is reduced [29].

#### 2.3.3. Sway Power Index

Fourier transform analysis is a mathematical representation of the wave signal of body vibrations in a horizontal plane made by the patient in order to maintain a vertical posture. When a body sway is detected on the ground reaction force sensors, various frequency components included in the measured value are divided into four frequency regions through a Fourier transformation of postural sway to calculate the sway power. The cause of increased body sway can be analyzed for each sensory organ by dividing the frequency as follows [30]:Low frequency region: This refers to the 0.01–0.1 Hz region, and an abnormally increased value is associated with a visual dysfunction.Mid/low frequency region: This refers to the 0.1–0.5 Hz region, and an abnormally increased value is associated with a disorder of the peripheral vestibular system.Mid/high frequency region: This refers to the 0.5–0.75 Hz region, and an abnormally increased value is associated with a somatosensory dysfunction.High frequency region: This refers to the 1.0–3.0 Hz region, and an abnormally increased value is associated with a disorder of the central nervous system.

### 2.4. Measurement Method

After an objective refraction test using a retinoscope (WelchAllyn, Auburn, NY, USA) and a 6 m LCD visual acuity chart (LUCID’LC, Everview, Seoul, Korea) by one experienced examiner, a subjective refraction test was performed with a manual phoropter (Ultramatic RX Master, Reichert, Depew, NY, USA) to detect the full correction value of each subject. Based on the detected full correction value, each subject was provided with glasses for distance. To measure the change in postural control ability, before and after wearing corrective glasses, the participants were asked to accurately align their bare feet with each ground reaction force sensor on the TETRAX. After 10 s of preparation time, in a static position, the data were collected by measuring for 32 s as instructed in the manual. The measurement order before and after wearing glasses was randomly determined for each individual in order to avoid effects caused by familiarity with the use of the device. During the measurement, the gaze target was set to look at the 0.1 numeric indicator which was fixed at 6 m in front in order to exclude adjustment. In consideration of the subject’s fatigue, a 5-min break was provided after the first measurement, followed by the second measurement. Based on the measured data, changes in the stability index, the synchronization index, and the sway power index were compared and analyzed for older adults who were wearing fully corrective glasses for distance.

### 2.5. Measurement Result Analysis

For data analysis, SPSS for Windows (Ver. 24.0, SPSS Inc, Chicago, IL, USA) was used, and paired *t*-test and Wilcoxon signed-rank test were performed in order to compare the averages of each evaluation factor before and after wearing corrective glasses. A statistically significant difference is determined in all analyses when the significance probability was *p* < 0.05.

## 3. Results

### 3.1. Changes in Stability Index

The changes in the stability index before and after correction with glasses in all 27 participants are shown in Figure 2. The average stability index was 25.59 ± 8.54 before correction with glasses and significantly decreased to 22.89 ± 6.64 (*t* = 2.544, *p* = 0.017) after correction with glasses, thereby indicating an overall improvement in the postural control ability of the elderly after wearing corrective glasses for distance. Table 2 shows the changes in the stability index after wearing corrective glasses for distance by classifying the 27 participants into older adults (younger than 80 years, 15 participants) and senior older adults (80 years or older, 12 participants) according to age. Based on the results, the average stability index in the older adult group was 25.74 ± 8.13 before correction with glasses and decreased to 24.37 ± 7.27 after correction with glasses (z = −0.538, *p* = 0.570), while the average stability index in the senior older adult group was 25.40 ± 9.39 before correction with glasses and 21.04 ± 5.50 after correction with glasses (z = −2.275, *p* = 0.023). The stability index decreased in both groups after wearing the corrective glasses compared to before the correction with glasses. However, a statistically significant difference was found only in the senior older adult group.

### 3.2. Changes in Synchronization Index

The changes in the six synchronization indices before and after wearing corrective glasses are shown in Figure 3. For AB (entire left foot), the average increased from –592.21 ± 339.64 before correction with glasses to –734.85 ± 255.60 after correction with glasses (*t* = 2.504, *p* = 0.019). For AC (left and right heels), the average increased from 511.44 ± 339.07 to 612.27 ± 286.13 (*t* = −2.176, *p* = 0.039). For BD (front parts of left and right soles), the average increased from 449.29 ± 375.26 to 632.98 ± 275.36 (*t* = −3.261, *p* = 0.003), and for BC (front part of left sole and right heel), the average increased from –803.33 ± 182.63 to –879.27 ± 80.66 (*t* = 2.258, *p* = 0.033). The four indices showed a statistically significant difference, but among the six synchronization indices, CD (entire right foot) and AD (left heel and front part of right sole) showed a tendency to increase after wearing corrective glasses. However, they did not show statistically significant differences.

### 3.3. Analysis of Sway Power Index

Table 3 presents the comparison and analysis of sway power in four frequency regions after wearing new corrective glasses. First, among the four frequency regions, measured before and after wearing corrective glasses, in all 27 participants, sway power significantly decreased only in the mid–high frequency region (*t* = 2.557, *p* = 0.017) and the high frequency region (*t* = 2.560, *p* = 0.017). The results of analyzing the sway power for each frequency region by classifying the participants into the older adult and senior older adult groups according to age are as follows: In the senior older adult group, sway power significantly decreased after wearing corrective glasses in the mid–high frequency region (z = −2.353, *p* = 0.019) and the high-frequency region (z = −2.432, *p* = 0.015). However, no clear change was found in the older adult group.

## 4. Discussion

A stable posture is maintained not only by visual information, but also by a complex inter-harmony of the sensory nervous system, including the vestibular and proprioceptive systems, and the motor nervous system responsible for muscle strength and reaction speed. As we age, the muscle strength responsible for postural control naturally declines and is known to decrease by approximately 40% by the age of 80 [31]. As shown in Figure 2, after wearing corrective glasses, the stability index of all participants significantly decreased, thereby indicating that the overall postural control ability was improved. Following the analysis of the changes in the stability index by classifying the participants into the older adult (66 to 79 years old) and senior older adult (80 years and older) groups (Table 2), the result showed that the positive effect of wearing corrective glasses was more evident in the senior older adult group than in the older adult group. Table 1 shows that the level of improvement in binocular visual acuity after wearing corrective glasses was similar in the two groups, with about two lines in the visual acuity chart on average (in the older adult group, from 0.69 for unaided visual acuity to 0.88 for corrective visual acuity; in the senior older adult group, from 0.51 for unaided visual acuity to 0.73 for corrective visual acuity). However, the average corrective visual acuity of the older adult group was higher than that of the senior older adult group. The results clearly indicate that the effect of improving postural control ability was more evident in the senior older adult group even though the older adult group obtained a higher corrected visual acuity after wearing corrective glasses. Additionally, the average spherical equivalent (SE) refractive power in the binocular was higher in the senior older adult group than in the older adult group (SE +0.31 ± 0.97 D in the older adult group, SE +0.72 ± 0.98 D in the senior older adult group). This difference in refractive error power in each group may lead to a result showing a significant change in the senior older adult group, only. Therefore, we suggest that the improved visual information resulting from refractive error correction provides a greater compensatory action to improve the overall postural control ability for the senior older adult group, whose posture maintenance ability is relatively more deteriorated. Anand et al. [32] reported that the risk of falls increased when refractive errors were experimentally induced by using spherical lenses in the elderly and that correcting uncorrected refractive errors is an important intervention strategy for preventing falls in the elderly. This study is considered to have great significance since it demonstrates the improvement effect of postural control ability in actual practice by having the participants wear corrective glasses rather than the experimental conditions as seen in the previous studies.

As previously mentioned, body balance ability is considered normal if the absolute value of the synchronization index is 700 or higher, whereas synchronization ability between each foot area is reduced if the absolute value of the synchronization index is lower than 700. However, the synchronization index is sometimes measured as an absolute value lower than 200 in the case of knee and ankle injuries or diseases of the cerebellum or cerebrum [29]. According to Lee and Eric [33], the main cause of increased fall risks and the occurrence of postural sway in the elderly is the functional deterioration of the foot and ankle joints. Meanwhile, Lee and Lishman [34] stated that as age increases, the ability to process information from the foot and ankle joints decreases, thus increasing the dependence on visual information for balancing. According to the results of this study, wearing corrective glasses alone showed a noticeable increase in four factors (AB, AC, BD, and BC) of the six synchronization indices, and synchronization ability entered the normal range of 700 or its boundary range (Figure 3). Kang [27] reported that four out of the six synchronization abilities improved after functional electrical stimulation training for stroke patients. Park and Kang [28] also reported that when visual biofeedback simulation training was applied to patients with incomplete spinal cord injuries, the overall synchronization ability was improved. The older adults who participated in this study had a little vision improvement effect after correction with glasses due to the effects of senile cataracts. However, it was confirmed that complete correction of the refractive errors in the elderly is a useful optical intervention to improve interaction and coordination in degenerated lower extremities. Our study results emphasize that optimal optical correction could potentially be an important factor in maximizing the therapeutic effect in various areas of rehabilitation treatment for older adults.

It is possible to determine the effect of visual correction on each sensory organ that contributes to posture control by analyzing the sway power index in a specific frequency region with the Fourier transform method provided by the TETRAX system.

Excessive sway in a specific frequency range is interpreted as a result of a pathological problem in the corresponding sensory organ or compensatory effort [35,36]. We attempted to investigate and analyze the influence and the cause of refractive error correction on posture control and synchronization ability in older adults. According to related clinical studies using the TETRAX system, Kollmitzer et al. [37] confirmed that the increase in sway power in the mid–high frequency region is a sign of somatic nervous system dysfunction related to lower extremity, spine, and back movements. DeWit [35] stated that an increase in sway in the high frequency region is often indicative of tremor-related central nervous system symptoms. Furthermore, it is interpreted as abnormalities in the cerebellum, cerebrum, and proprioception. In a previous study [24], we reported that sway power decreased only in the mid–high frequency region when actual fully corrective glasses were fabricated and worn by healthy young participants with myopic refractive error. Based on this, we have proven that optical correction of myopic refractive error has a positive effect on the somatic nervous system among sensory organs that contribute to posture control. As shown in Table 3, in the 27 older adults participating in this study, among the four frequency regions, the sway power index showed a statistically significant decrease, in the mid–high frequency region related to the somatic nervous system function and the high frequency region related to the sensory integration ability of the central nervous system, after wearing corrective glasses compared to before wearing them. These results were found due to significant changes in the senior older adult group. Although the prescription of fully corrective glasses did not achieve dramatic visual acuity improvement, it did effect the improvement of the sensory integration function of the central nervous system that controls the posture of the elderly and the lower extremity motor ability based on the somatic nervous system. As a result, it was determined that changes in these sensory organs through refractive error correction contributed to improving the postural stability and synchronization ability of the older adults who participated in this study. According to Woollacott et al. [38], healthy elderly people can exhibit postural control ability similar to that of the younger people in situations where the effectiveness or accuracy of only one of the senses required for postural control is impaired. However, in contrast to the young participants, when the effectiveness of the two sensory information systems was lowered, postural instability clearly increased even in healthy elderly people. Therefore, providing optimal visual information by correcting residual refractive error should be prioritized, because it can help older adults, with other degenerated sensory functions, to maximize and maintain postural control ability. Finally, based on the results of this study, we suggest that the accurate correction of refractive errors can be used as an optical intervention strategy to prevent the risk of falls among older adults worldwide.

Our study had the following limitations: First, there was a limitation in recruiting older participants who, within the last year, had no experience wearing corrective glasses for distance and who had not previously undergone the surgery or treatment for eye diseases such as cataracts, glaucoma, macular degeneration, retinal problems caused by diabetes, hypertensive retinal disease, and strabismus. For this reason, a sufficient number of participants could not be secured. Furthermore, further research is required to verify the effect of optical correction for refractive error on postural control ability in patients with various eye diseases. Second, since this study focused on whether the wearing of corrective glasses has a positive effect on the postural control ability in older adults and its cause, correlation analysis was not performed between the postural control ability and visual function (visual acuity level, refractive error level, contrast sensitivity ability, stereopsis, etc.). Third, since the results of this study were analyzed immediately after the corrective glasses were worn based on results measured for 32 s, the adaptation phenomenon after the corrective glasses were worn was not considered. A follow-up study needs to be conducted in order to address the aforementioned limitations.

## 5. Conclusions

In this study, the positive effect of corrective glasses on postural control ability was verified. We studied 27 older adults in actual practice, who had not worn corrective glasses for more than a year, by fitting them with corrective glasses. We demonstrated that wearing corrective glasses improved the synchronization ability of each foot of older adults, as well as their overall postural stability. It was determined that the accuracy of visual information through refractive error correction led to the improvement of the somatic nervous system and sensory integration ability among the sensory organs responsible for posture control. This effect was more pronounced in the senior older adult group (aged 80 years or older). Fall incidence among the elderly is a global concern, as it is highly detrimental to their health; each specialized medical field should therefore prioritize the prevention of fall incidence. For this reason, we emphasize the importance of addressing uncorrected refractive error in the specialized field of optometry and suggest that it might be the most basic and essential strategy for improving the postural control ability of older adults and preventing fall accidents. It is our hope that this study will be utilized as a reference to emphasize the social role of optometrists and to be considered in related specialized fields such as orthopedic surgery, physical therapy, and occupational therapy.

## Figures and Tables

**Figure 1 ijerph-19-06643-f001:**
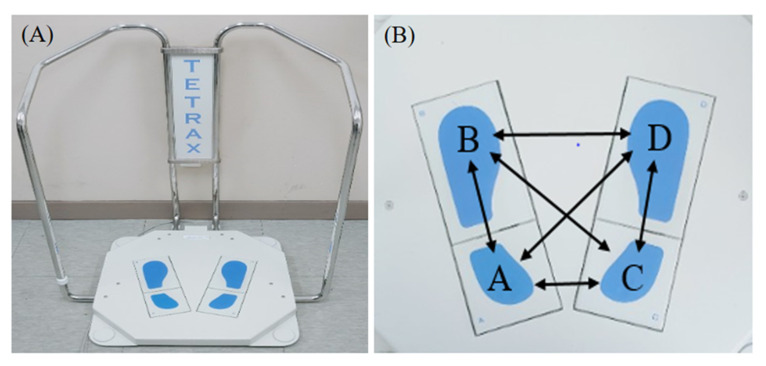
TETRAX postural balance device (**A**) and schematic diagram of six synchronizations on four plates (**B**).

**Figure 2 ijerph-19-06643-f002:**
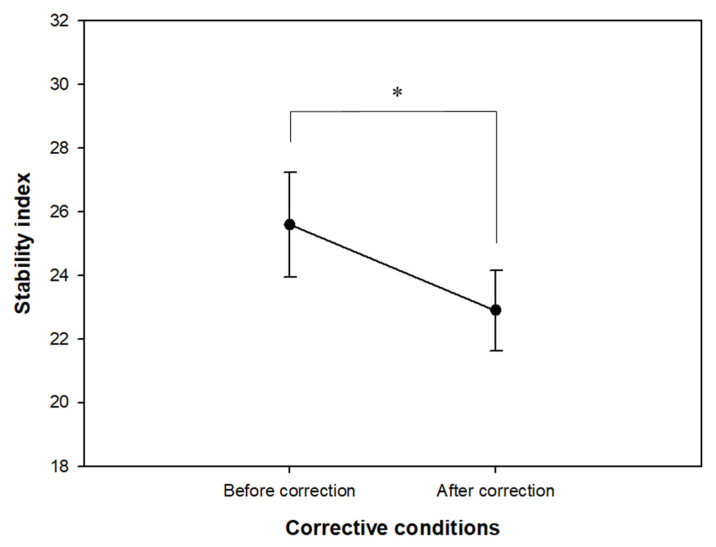
Changes in stability index before and after wearing corrective glasses in all participants (*n* = 27). * *p* < 0.05 by paired *t*-test.

**Figure 3 ijerph-19-06643-f003:**
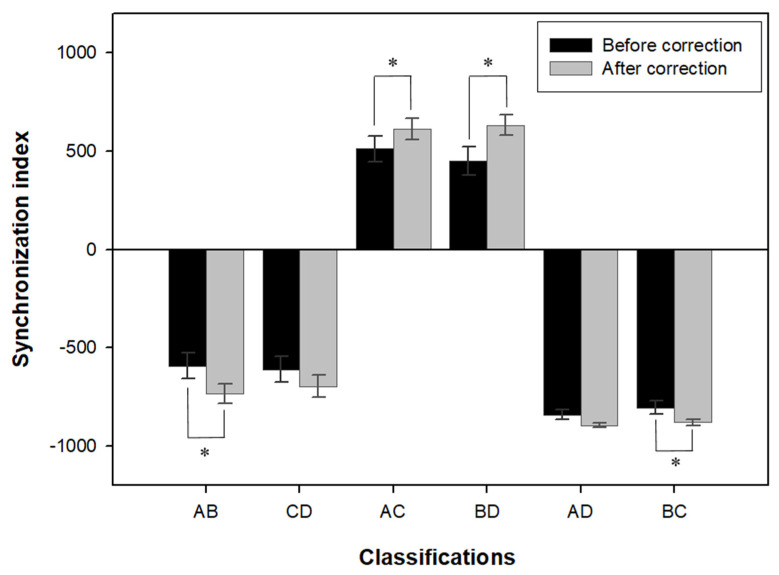
Changes in synchronization index before and after wearing corrective glasses in all participants (*n* = 27). AB: entire left foot; CD: entire right foot; AC: left and right heels; BD: front parts of left and right soles; AD: left heel and front part of right sole; BC: front part of left sole and right heel. * *p* < 0.05 by paired *t*-test.

**Table 1 ijerph-19-06643-t001:** Information of full corrective prescription and visual acuity in all participants.

No	Age	Full-Corrective Prescription (Unit: Diopters)		Decimal Visual Acuity					
		Right Eye	Left Eye	Before Correction			After Correction		
				RE ^1^	LE ^2^	BE ^3^	RE	LE	BE
1	87	S+1.50 C−0.75 Ax140	S+2.25 C−1.75 Ax55	0.3	0.5	0.6	0.6	0.7	0.8
2	84	S+1.00 C−0.50 Ax90	S+1.00 C−0.50 Ax90	0.5	0.5	0.6	0.7	0.8	0.8
3	78	S+1.50 C−0.50 Ax100	S+0.75 C−0.50 Ax90	0.3	0.5	0.6	0.6	0.7	0.8
4	80	S+1.75 C−0.75 Ax70	S+2.00 C−0.75 Ax85	0.5	0.5	0.7	0.8	0.8	0.9
5	86	S+0.50 C−1.00 Ax110	S+1.00 C−1.00 Ax80	0.3	0.2	0.4	0.4	0.5	0.6
6	89	S−0.25 C−1.00 Ax135	C−0.50 Ax135	0.2	0.4	0.6	0.8	0.7	0.8
7	85	S+2.50 C−0.50 Ax155	S+2.75 C−0.75 Ax105	0.2	0.2	0.4	0.5	0.5	0.7
8	90	S+1.75 C−1.50 Ax90	S+2.50 C−1.75 Ax65	0.1	0.2	0.2	0.3	0.3	0.5
9	86	C−2.00 Ax105	S+0.50 C−2.00 Ax60	0.2	0.2	0.4	0.4	0.5	0.6
10	81	S−0.50	S+1.00 C−0.75 Ax90	0.4	0.5	0.6	0.7	0.7	0.8
11	81	S+1.00 C−1.50 Ax100	S+1.50 C−1.75 Ax110	0.6	0.5	0.7	0.7	0.8	0.9
12	90	S+2.00 C−0.75 Ax95	S+2.00 C−0.75 Ax70	0.1	0.1	0.3	0.4	0.4	0.5
13	68	S+0.50 C−1.00 Ax90	S+1.00 C−1.00 Ax80	0.4	0.3	0.6	0.8	0.7	0.9
14	77	C−1.25 Ax180	C−1.00 Ax165	0.4	0.6	0.7	0.7	0.8	0.9
15	75	S+0.50 C−0.50 Ax170	S+1.75 C−0.25 Ax180	0.5	0.4	0.6	0.9	0.8	0.9
16	75	S+2.00 C−2.00Ax75	S+1.25 C−1.50 Ax110	0.2	0.4	0.5	0.5	0.7	0.8
17	79	S+1.75 C−1.00 Ax60	S+1.25 C−0.75 Ax130	0.1	0.1	0.2	0.6	0.6	0.7
18	74	S+2.00 C−1.00 Ax90	S+3.25 C−0.75 Ax115	0.5	0.3	0.7	0.6	0.6	0.8
19	69	S−0.50 C−2.50 Ax95	S+0.75 C−2.00 Ax80	0.3	0.6	0.6	0.8	0.7	0.8
20	78	S+2.00 C−0.75 Ax100	S+2.25 C−1.00 Ax90	0.3	0.3	0.5	0.6	0.6	0.8
21	71	S−0.50	S+0.25 C−1.00 Ax130	0.7	0.8	0.9	1.0	1.0	1.0
22	77	S+1.00	C−0.50 Ax105	1.0	0.8	1.0	1.0	1.0	1.0
23	79	S+0.50 C−0.75 Ax90	S+1.25 C−2.00Ax70	0.7	0.6	0.7	0.7	0.7	0.8
24	66	S+0.50	S+0.50 C−0.50 Ax75	1.2	1.0	1.2	1.2	1.2	1.2
25	78	S−0.75 C−0.50 Ax170	C−0.50 Ax60	0.5	0.7	0.7	0.8	0.8	0.8
26	77	S+0.75 C−1.75 Ax105	S+0.50 C−1.50 Ax 90	0.4	0.4	0.6	0.7	0.6	0.8
27	69	S−0.25	S−0.50	0.6	0.5	0.8	1.0	0.9	1.0

^1^ Right eye; ^2^ left eye; ^3^ both eyes.

**Table 2 ijerph-19-06643-t002:** Changes in stability index before and after wearing corrective glasses according to age groups.

Age Groups (*n*)	Correction Condition	Stability Index	*p*-Value
Total (27)	Before	25.59 ± 8.54	*t* = 2.254, *p* = 0.017 *
	After	22.89 ± 6.64	
≥80 (12)	Before	25.40 ± 9.39	z = −2.275, *p* = 0.023 **
	After	21.04 ± 5.50	
80 > (15)	Before	25.74 ± 8.13	z = −0.568, *p* = 0.570
	After	24.37 ± 7.27	

* *p* < 0.05 by paired *t*-test, ** *p* < 0.05 by Wilcoxon test.

**Table 3 ijerph-19-06643-t003:** Changes in sway power index in each frequency range before and after wearing corrective glasses.

Age Groups (*n*)	Correction Condition	Sway Power Index in Each Frequency Range			
		Low	Low to Medium	Medium to High	High
Total (27)	Before	21.99 ± 9.05	10.58 ± 3.04	4.90 ± 1.83	0.83 ± 0.29
	After	20.47 ± 7.11	10.73 ± 3.31	4.23 ± 1.54	0.73 ± 0.26
	*t*/*p*-values	0.919/0.367	0.296/0.769	2.557/0.017 *	2.560/0.017 *
≥80 (12)	Before	20.71 ± 7.19	10.51 ± 3.81	4.79 ± 1.64	0.83 ± 0.24
	After	19.43 ± 6.89	10.53 ± 3.67	3.76 ± 1.20	0.67 ± 0.21
	z/*p*-values	−0.392/0.695	−0.157/0.875	−2.353/0.019 **	−2.432/0.015 **
80 > (15)	Before	23.00 ± 10.44	10.64 ± 2.39	4.99 ± 2.02	0.84 ± 0.33
	After	21.31 ± 7.41	10.89 ± 3.11	4.60 ± 1.71	0.78 ± 0.29
	z/*p*-values	0.000/1.000	−0.682/0.496	−0.682/0.427	−0.511/−0.609

* *p* < 0.05 by paired *t*-test, ** *p* < 0.05 by Wilcoxon test.

## Data Availability

The data presented in this study are available upon request from the corresponding author.

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
