# Peer review of "Changes in Postural Control Ability after Wearing Corrective Glasses for Distance in Older Adults and Their Causes"

_ijerph, 2022, doi:10.3390/ijerph19116643_

Round 1
Reviewer 1 Report
The manuscript is well written and it covers all the aspects of an acceptable paper to be published.
just minor comments:
In Table 1 we need to know the visual acuity of each eye alone and then the visual acuity of both eyes together,
2-All glass prescriptions are hypermetropic astigmatic and non shows myopic errors, so this should be clarified by the author,
3- the posture does not only depend on the visual acuity but also on the field of vision , so patients with glaucoma or other ocular diseases should be explained and excluded and clarified,
4- also the author didn't explain how many of these patients had surgeries in their eyes like cataract, glaucoma or retinal detachment.
5- another very important point is the central vision , reading capabilities, if these patients have a macular problems due to age or diabetes.
6- Diabetics should be clarified for their diabetic retinopathy and maculopathy status
Author Response
Response to Reviewer 1 Comments
The manuscript is well written and it covers all the aspects of an acceptable paper to be published.
Response : We thank the reviewer for the positive critique.
Point 1: In Table 1 we need to know the visual acuity of each eye alone and then the visual acuity of both eyes together,
Response 1: We thank the reviewer for the positive critique. As pointed out by the reviewers, Table 1 was modified by adding the monocular uncorrected visual acuity and monocular corrected visual acuity of each subject (Line 104-106).
Point 2: All glass prescriptions are hypermetropic astigmatic and non shows myopic errors, so this should be clarified by the author,
Response 2: Thank you for your detailed commects. Three patients only had mild myopic refractive error in both eyes among the 27 subjects included in this study. We do not specifically limit the type of refractive error. The reason is that because the experience of wearing glasses for distance can interfere with the analysis results. So, our study was limited to subjects who did not have experience with wearing glasses for distance within the last 1 year. For this reason, it is considered that the type of refractive error of the subject is mainly hypermetropic, additionally, due to the change toword hyperopic with increasing age. We added the explanations for this issue in “participants” section as follows (Line 85-88).
“In the 27 subjects included in this study, 18 subjects had hyperopic refractive error in both eyes, 3 subjects had myopic refractive error in both eyes, and 6 subjects had a combination of hyperopic and myopic refractive error in each eye.”
Point 3: the posture does not only depend on the visual acuity but also on the field of vision, so patients with glaucoma or other ocular diseases should be explained and excluded and clarified,
Response 3: Thank you for your detailed commects. We agree with the reviewer's commnets. In studies related to postural control and vision, there is a previous report that peripheral vision acts as a factor affecting postural stability. The reason why this study only specified subjects without a history of systemic and eye diseases surgery or treatment is that the corrective effect of glasses can be affected by various such diseases.
Please note that we met nearly 100 subjects over the path year, and we finally selected 27 subjects for this study after excluding subjects who have had surgery or treatment experience for various systemic and eye diseases. Therefore, we consider that there were no subjects in this study with peripheral visual field problem due to glaucoma or other diseases as you are concerned. We added the explanations for this issue in “participants” section as follows (Line 91-93).
“Medical interviews confirmed that the participants had no history of glaucoma, macular degeneration, retinal problem caused by diabetes, hypertensive retinal disease, strabismus and experience in surgery or treatment related thereto, and frequent falls or any diseases related to body imbalance, systemic diseases, and medications (neuromuscular and musculoskeletal).”
Point 4: also the author didn't explain how many of these patients had surgeries in their eyes like cataract, glaucoma or retinal detachment.
Response 4: Thank you for your detailed commects. We acknowledge that the detailed explanation of the selection of subjects was insufficient. As answered point 3, we have revised including this issue in “participants” section as follows (Line 91-93).
Point 5: another very important point is the central vision, reading capabilities, if these patients have a macular problems due to age or diabetes.
Response 5: As I explained about the selection of subjects, we think carefully that this may be not related to this issue because the subjects of this study do not have macular problems due to diabetes. It is also emphasized that there were no patients complaining of cognitive difficulties or reading difficulties during the examination due to age, although the super-aged subjects exist in this study.
Most of the subjects in this study had a corrected visual acuity of 1.0 or less. We think that these results are due to degenerative changes due to age, and these results are not necessarily related to eye diseases and can also be thought of as visual functional deterioration.We added the explanations for this issue in “participants” section based on your advice as follows (Line 97-99)
“In addition, there were no subjects with impaired reading ability or cognitive problems during the visual acuity and refraction tests due to the elderly.”
Point 6: Diabetics should be clarified for their diabetic retinopathy and maculopathy status
Response 6: We confirmed that there were no subjects who had diabetics by Medical interviews. Through questions 3 to 6 advised by the reviewers, we are going to investigate the effect of corrective glasses on postural stability depending on various eye diseases in collaboration with ophthalmologist and orthopedic specialists in the next experiment. Your advice will lead to our next work and we really appreciate it. We have revised the limitation of this study in “Discussion” section based on your advice as follows (Line 337-342).
“Our study had the following limitations. First, there was a limitation recruiting older participants who, within the last year, had no experience wearing corrective glasses for distance and who had not previously undergone the surgery or treatment for eye diseases such as cataract, glaucoma, macular degeneration, retinal problem caused by diabetes, hypertensive retinal disease, strabismus. For this reason, a sufficient number of participants could not be secured. Furthermore, further research is required to verify the effect of optical correction for refractive error on postural control ability in patients with various eye diseases.
Reviewer 2 Report
The reviewer feel that the manuscript is ready for publication but suggest that you will include the answer regarding the experimental procedure and average corrective error in the text, so the reader will understand better this paper.
1. The experiment was performed without correction and then after correction. Don’t the authors think it could affects the results? Because in the first experiment, the participants could become familiar with the use of the device.
2. It would be better to present the average corrective error in older adults and senior older adults to understand the results. Because a statistically significant difference was found only in the senior-older adult group. And the authors need to add some explanation.
Author Response
Response to Reviewer 2 Comments
The reviewer feel that the manuscript is ready for publication but suggest that you will include the answer regarding the experimental procedure and average corrective error in the text, so the reader will understand better this paper.
Response : We thank the reviewer for the positive critique.
Point 1: The experiment was performed without correction and then after correction. Don’t the authors think it could affects the results? Because in the first experiment, the participants could become familiar with the use of the device.
Response 1 : I totally agree with the reviewers' comments. We randomly performed the measurement sequence before and after wearing glasses to prevent errors due to repeated measurements. This issue has already been mentioned on line 168-169 (”The measurement order before and after wearing glasses was randomly determined for each individual).
To enhance the reader's understanding, we added the explanations for this issue as follows (Line 169-170)
“The measurement order before and after wearing glasses was randomly determined for each individual in order to avoid effects caused by familiarity with the use of the device.”
Point 2: It would be better to present the average corrective error in older adults and senior older adults to understand the results. Because a statistically significant difference was found only in the senior-older adult group. And the authors need to add some explanation.
Response 2 : Thank you for your insightful comments and prospective suggestions. We added the explanations for this issue in “Discussion” section based on your advice, as follows (Line 257-262).
“Additionally, the average of spherical equivalent (SE) refractive power in the binocular was higher the senior-older adult group than in the older adult group (SE +0.31±0.97 D in the older adult group, SE +0.72±0.98 D in senior-older adult group, respectively). This difference in refractive error power in each group may lead to a result showing a significant change in the senior-older adult group, only.”
